# 30-day unplanned readmission rate in otolaryngology patients: A population-based study in Thuringia, Germany

**Wido Rippe[1], Andreas Dittberner[1], Daniel Boeger[2], Jens Buentzel[3], Kerstin Hoffmann[4], Holger Kaftan[5], Andreas Mueller[6], Gerald Radtke[7], Orlando Guntinas-Lichius[1]***

**1** Department of Otorhinolaryngology, Jena University Hospital, Jena, Germany, **2** Department of Otorhinolaryngology, SRH Zentralklinikum, Suhl, Germany, **3** Department of Otorhinolaryngology, Südharz-Krankenhaus gGmbH, Nordhausen, Germany, **4** Department of Otorhinolaryngology, Sophien/Hufeland-Klinikum, Weimar, Germany, **5** Department of Otorhinolaryngology, HELIOS-Klinikum, Erfurt, Germany, **6** Department of Otorhinolaryngology, SRH Wald-Klinikum, Gera, Germany, **7** Department of Otorhinolaryngology, Ilm-Kreis-Kliniken, Arnstadt, Germany

* orlando.guntinas@med.uni-jena.de

**Data Availability Statement:** Nearly all data are presented in the manuscript. Due to ethical restrictions on data sharing related to participant consent, aggregated data are available upon

## Abstract

### Purpose

Analyze associations between patients' characteristics and treatment factors with 30-day unplanned readmissions in hospitalized otolaryngology patients in the German Diagnosis Related Group (D-DRG) system.

### Methods

A retrospective cohort study was performed on 15.271 otolaryngology admissions of 12.859 patients in 2015 in Thuringia, Germany. The medical records of the 1173 cases (7.6%) with readmission within 30-days were analyzed in detail.

### Results

The 30-day readmission was planned in 747 cases (4.9%) and was unplanned in 422 cases (2.8%). The median interval between primary and next inpatient treatment was 11 days. The principal diagnosis was the same as during the primary index treatment in 72% of the cases. The most frequent reasons for readmission were: Need for non-surgical therapy (31.2%), need for further surgery (26.3%), post-surgical complaints (16.9%), and recurrence of primary complaints (10.7%). The multivariate analysis revealed that discharge due to patient's request against medical advice was a strong independent factor with high risk for unplanned readmission (Odds Ratio [OR] = 9.62]; confidence interval [CI] = 2.69–34.48). Surgery at index admission (OR = 3.33; CI = 1.86–5.96) was the second important independent risk factor for unplanned readmission. Unplanned readmission had more frequently a non-surgical treatment at readmission than a surgical treatment (OR = 3.92; CI = 2.24–6.84) and needed more frequently further diagnostics (OR = 2.34; CI = 1.34–4.11). The following index International Classification of Diseases (ICD) categories had the highest risk for unplanned readmission: Injury, poisoning and certain other consequences of external

request to the ENT department, University Hospital Jena, D-07747 Jena, Germany (hno@krz.uni-jena. de).

**Funding:** The authors received no specific funding for this work.

**Competing interests:** The authors have declared that no competing interests exist.

causes, ICD: S00-T98 (OR = 66.67; CI = 15.87–333.33), symptoms, signs, abnormal findings, ill-defined causes, not otherwise classified, ICD: R00-R99 (OR = 62.5; CI = 11.76–333.33), blood forming organ diseases, ICD: D50-D90 (OR = 21.276; CI = 3.508–125), and eye/ ear diseases, ICD: H00-H95 (OR = 12.66; CI = 4.29–37.03).

## Conclusions

The causes of unplanned 30-day readmission in German otolaryngology inpatients are multifactorial. Specific patient and treatment characteristics were identified to be targeted with health care interventions to decrease unplanned readmissions.

## Introduction

Unplanned hospital readmissions are an outcome measure in health services research as metric for health care quality and are costly [1]. Potentially avoidable readmissions can be the consequence of an adverse event or a too early discharge of a prior hospitalization [2]. One of the most widely used tools for reimbursing inpatient health services around the world is Diagnosis-Related Groups (DRGs) [3]. The risks of early discharge in order to cut costs have been well documented ever since DRG systems were first introduced [4]. Therefore, instruments were applied to include hospital readmission rates in reimbursement decisions. In the United States, for instance, the Hospital Readmission Reduction program (HRRP), a Medicare value-based purchasing program was introduced with the aim to reduce payments to hospitals with excess readmissions [5]. In the German DRG (G-DRG) system, readmissions for the same cause within 30 days after discharge are reimbursed by the original DRG and receive no additional funds. This approach financially penalizes inappropriate early discharge (at least if it leads to readmission) [4].

These two programs as many others use a time frame within 30 days of discharge, i.e. the 30-day readmission rate as a parameter because it is said that readmissions during this time can be influenced by the quality of care received at the hospital and how well discharges were coordinated. Later readmissions may not be related to the primary (index) inpatient care. Later readmissions might be more related to the outpatient care the patient receives. Other later influencing factors might be individual health choices and behaviors, and community-level factors beyond the control of the hospital that treated the patient first.

Only a few studies, which are mainly based on the experience in a single hospital, have been performed to analyze risk factors for unplanned hospital readmission in otolaryngology patients [6–8]. A larger population-based analysis has only be performed for head and neck cancer surgery cases using the American College of Surgeons National Surgical Quality Improvement Program (NSQIP) database [9]. Population-based analyses for non-surgical otolaryngology inpatients and any data for Germany are lacking so far.

Thuringia is a territorial state in Germany with approximately 2.2 million habitants. There are only eight hospitals with departments of otolaryngology. The departments of otolaryngology have built a network primarily to improve health services research in the field of otolaryngology (for instance, [10–13]). Use of this network provided an ideal platform for a population-based analysis of the 30-day readmission rates of unselected otolaryngology inpatients treated in 2015 in Thuringia in daily practice with focus on unplanned readmissions and its predictors.

## Material and methods

A standardized retrospective analysis was performed in seven Thuringian hospitals that have a department of otolaryngology (the eighth hospital did not take part). These seven hospitals cover about 90% of all inpatient otolaryngology cases in Thuringia. The institutional ethics committee (Ethics Committee of the University Hospital Jena, Germany) approved the study protocol. The ethics committee waived the requirement for informed consent for the patients with exclusive retrospective data analysis using the patients' charts.

All otolaryngology patients with inpatient treatment in 2015 were included. Outpatients and day-care patients were excluded. The patients were identified via the hospital information systems of the seven participating hospitals. As part of the G-DRG system and according to the Section 21 Hospital Remuneration Act (Krankenhausentgeltgesetz; KHEntgG), the hospitals have to prepare standardized datasets on patients' characteristics including Patient Clinical Complexity (PCCL) coding, International Statistical Classification Of Diseases And Related Health Problems, 10th revision, German Modification coding (ICD-10-GM), and German Operations and Procedures Key (OPS) coding. The patient ID was used to identify all patients with readmission within 30 days. These patients built the group of primary interest (readmission group). The other patients built the group of patients without 30-day readmission (no readmission group). To characterize the readmission group in more detail, and especially to identify the patients with planned versus unplanned readmission, the patients' charts of all patients with 30-day readmission were revisited. A planned readmission was defined as a readmission that was planned during the index admission. For instance, a patient with head and neck cancer had his index admission for staging and panendoscopy. Typically, if surgery was planned for definitive treatment, the day of readmission already was scheduled before demission. All other patients without planned readmission were defined as unplanned readmissions, for instance when a patient was readmitted for a complication after surgery.

The primary aim of the subsequent evaluation was to analyze associations between patients' characteristics and treatment factors with 30-day unplanned readmissions in hospitalized otolaryngology patients in the German Diagnosis Related Group (D-DRG) system. The G-DRG system already includes instruments to avoid early readmission, especially by sanctioning the reimbursement. We hypothesized that, nevertheless, specific otolaryngology diseases and type of treatment as well as patients' comorbidity have influence on the risk of 30-day unplanned readmission.

## Statistical analysis

Patient demographics and outcome variables were analyzed with IBM SPSS statistics software (IBM Corp, released 2017, IBM SPSS Statistics for Windows, Version 25.0. Armonk, New York). Data are presented as frequencies or mean ± standard deviation (SD) if not otherwise indicated. To compare the no readmission group with the readmission group and to compare the planned readmission subgroup with the unplanned readmission subgroup, ordinal and nominal data were compared with the chi-square test. Scaled data were compared with the non-parametric Mann-Whitney U-test. The significance level was set at $p < 0.05$. Factors with significant differences between groups were included in the multivariate analysis: Binary logistic regression analysis was used to determine independent factors for 30-day readmission and for unplanned readmission.

## Results

### Comparison of patients without and with 30-day readmission

15271 inpatient cases of 12859 different patients were registered in 2015. 12925 cases had no 30-day readmission. The remaining 2346 cases equally consisted of 1173 primary cases and the

1173 related cases of 30-day readmission. An overview about all patients is given in **S1 Table**. The localizations of the diseases and the hundred most frequent ICD codes are presented in **S1 Fig** and **S2 Fig**, respectively. The univariate comparison between patients without and with 30-day readmission is summarized in **S2 Table**. Patients with readmission were older ($p<0.001$) and more frequently male ($p<0.001$). The primary inpatient treatment was longer in readmitted patients and these patients were treated in hospitals with higher volume ($p<0.001$, respectively). Patients with malignant disease had a higher risk of readmission ($p<0.001$). Patients with higher PCCL and higher comorbidity had a higher risk of readmission ($p<0.001$, respectively). Non-surgical cases had a higher risk of readmission than surgical cases ($p<0.001$).

According to the multivariate analysis (**S3 Table**) a patient with malignant disease had the highest risk of readmission (Odds ratio [OR] = 5.56; confidence interval [CI] = 4.35–7.14) followed by patients with high PCCL (OR = 2.05; CI = 1.65–2.54). Other relevant independent risk factors were male gender (OR = 1.20; CI = 1.01–1.41), higher number of secondary diagnoses (CI = 1.03; CI = 1.01–1.04), a non-surgical treatment (OR = 1.40; CI = 1.18–2.54), and a treatment in a hospital with higher volume (OR = 1.43; CI = 1.20–1.69).

## Comparison of patients with planned and unplanned 30-day readmission

1173 cases of 30-day readmission occurred, i.e. the 30-day readmission rate was 7.6%. The 30-day readmission rate for surgical cases only was 6.5%. Reasons for 30-day readmission were in descending order: Need for non-surgical therapy (31.2%), need for further surgery (26.3%), post-surgical complaints (16.9%), recurrence of primary complaints (10.7%), other complaints (9.9%), need for further diagnostics (4.7%), and reason unknown (0.2%). The principal diagnosis was the same as during the primary treatment in 72% of the cases. 37 cases were released from the hospital during the primary treatment earlier than planned against medical advice at the patient's request. The 30-day readmission was planned in 747 cases (4.9%) and was unplanned in 422 cases (2.8%). Primary localization of the disease of the planned and unplanned readmissions is shown in **Fig 1**. The oral cavity and the pharynx were the localization with the highest number of planned and unplanned 30-day readmissions. The absolute number of planned readmissions was also high in descending order for the localizations larynx/thyroid, neck, ear and face. The absolute number of unplanned readmission was higher than planned readmission in descending order for the localizations ear, paranasal sinus, nose, and other localizations. The fifty most frequent ICD codes at primary index admission of planned and unplanned readmissions are shown in **Fig 2**. The ten most frequent ICD codes for planned 30-day readmission belonged to the group of head and neck cancer codes. In contrast, only 3 ICD codes for head and neck cancer are found in the group of patients with unplanned readmission. More frequent were infectious diseases as index disease (chronic tonsillitis, rhinitis, sinusitis, peritonsillar abscess), and bleeding (mainly after oropharyngeal surgery). The interval between primary and next inpatient treatment was 12.5±8.2 days (median: 11). The treatment duration at readmission was 7.0±8.2 days.

The univariate comparison between patients with planned versus unplanned 30-day readmission is summarized in **Table 1**. Further surgery took place more often planned than unplanned ($p<0.001$), but further surgery because of postoperative complaints occurred more often unplanned ($p<0.001$). Recurrence of the original complaints was more frequently a reason for unplanned readmission ($p<0.001$). Patients with unplanned readmission were younger and more frequently female ($p<0.001$; respectively). The interval to planned readmission was longer than to unplanned readmission ($p<0.001$). The duration of the first inpatient treatment was longer if an unplanned readmission occurred ($p<0.001$). The PCCL and comorbidity

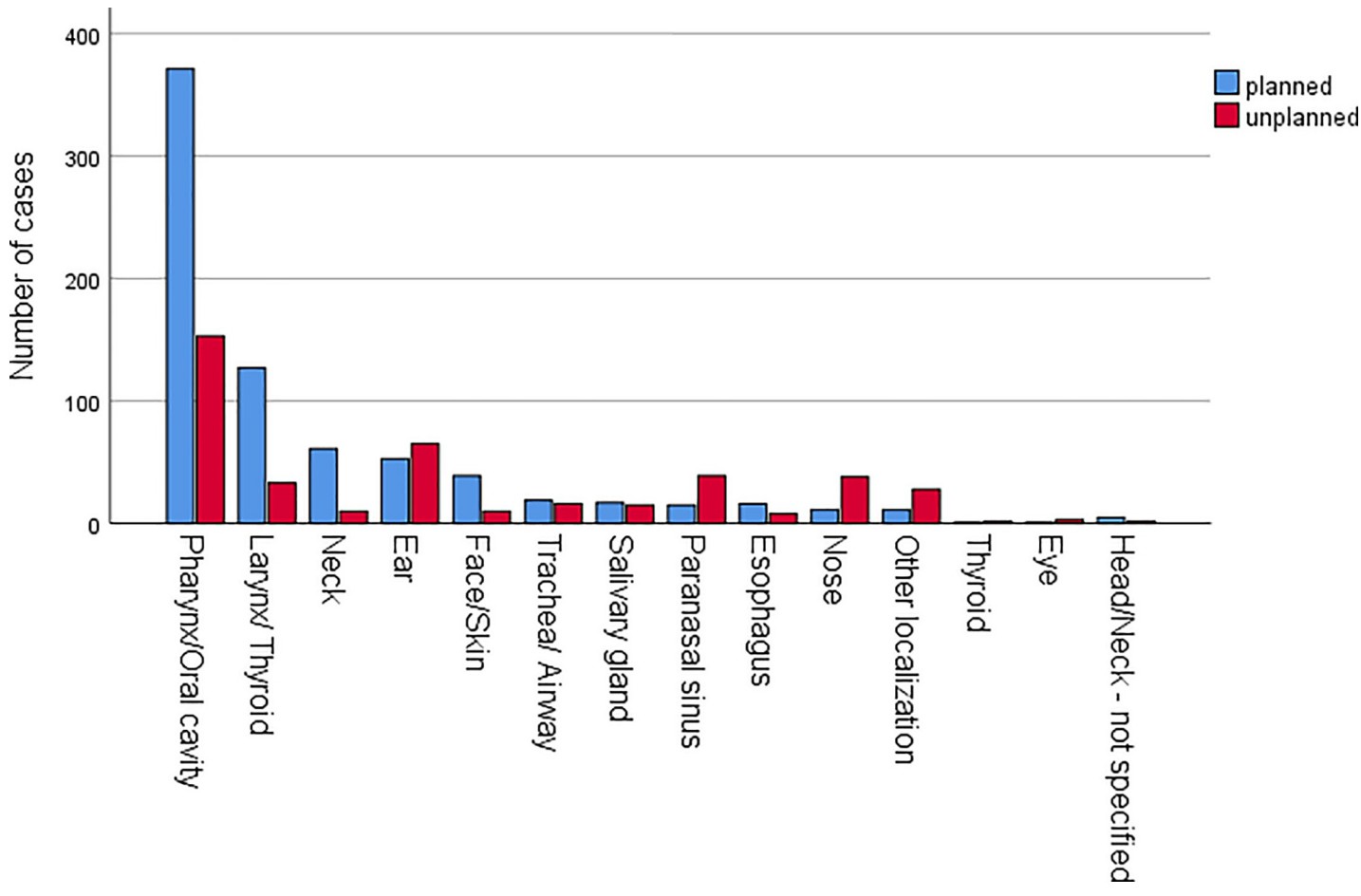

**Fig 1.** Localization of the primary disease in patients with planned (blue) and unplanned (red) 30-day readmission.

were higher for planned readmissions (p<0.001 respectively). Unplanned readmission was more frequently seen in patients who were discharged against medical advice during the prior inpatient treatment. Primary diseases with more unplanned readmission were: Certain infectious and parasitic diseases, ICD: A00-B99, ICD: A00-B99 (p<0.001), blood forming organ diseases, ICD: D50-D90 (p = 0.002), eye/ ear diseases, ICD: H00-H95 (p<0.001), circulatory system diseases, ICD: I00-I99 (p = 0.001), respiratory system diseases, ICD: J00-J99 (p<0.001), gastrointestinal tract diseases, ICD: K00-K93 (p<0.001), symptoms, signs, abnormal findings, ill-defined causes, not otherwise classified, ICD: R00-R99 (p<0.001) and injury, poisoning and certain other consequences of external causes, ICD: S00-T98 (p<0.001). Diseases with more planned readmission were: Malignant diseases, ICD: C00-C97 (p<0.001), and benign, in-situ, uncertain neoplasm, ICD: D00-D48 (p = 0.042).

The multivariate analysis is shown in **Table 2**. The most important predictors for 30-day unplanned readmission were: discharge due to patient's request against medical advice, surgical cases, and several ICD categories. Discharge due to patient's request against medical advice was a strong independent factor with high risk for unplanned readmission (OR = 9.62; CI = 2.69–34.48). Lower number of secondary diagnoses (OR = 1.17; CI = 1.08–1.27) were other independent risk factors for unplanned compared to planned readmission. Unplanned readmission had more frequently a non-surgical treatment at readmission than a surgical treatment (OR = 3.92; CI = 2.24–6.84) and needed more frequently further diagnostics

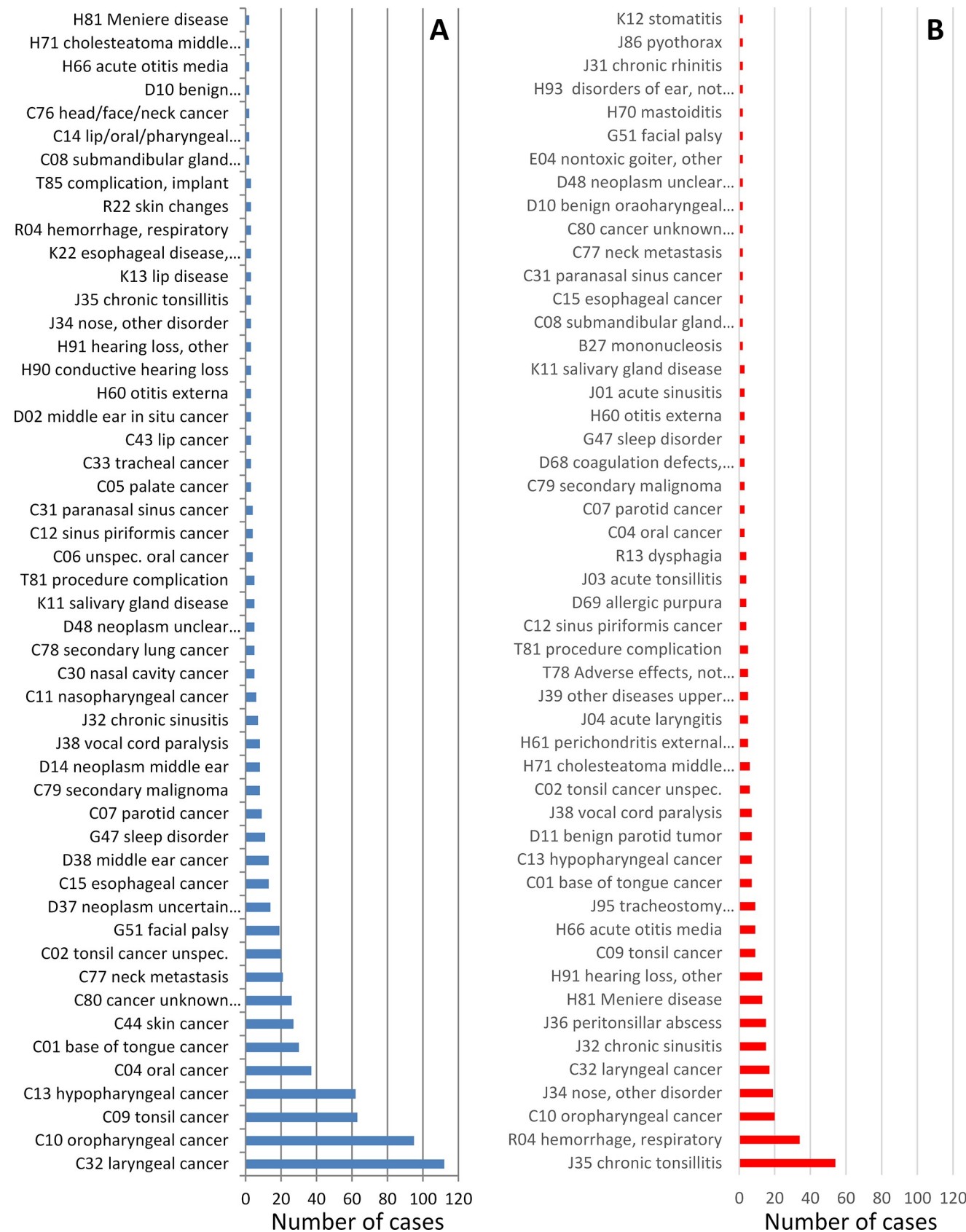

**Fig 2.** The fifty most frequent ICD codes of patients with 30-day readmission. A: planned (blue) readmission. B: Unplanned (red) readmission.

**Table 1. Comparison of the group of patients with planned versus unplanned 30-day readmission.**

| Parameter | All | Planned readmission | | Unplanned readmission | | p |
|---|---|---|---|---|---|---|
| | N | N | % | N | % | |
| **All** | 1169 | 747 | 63.9 | 422 | 36.1 | |
| **Gender** | | | | | | <0.001 |
| Male | 865 | 604 | 69.8 | 261 | 30.2 | |
| Female | 304 | 143 | 47.0 | 161 | 53.0 | |
| **Reason for readmission** | | | | | | |
| Further surgery | | | | | | <0.001 |
| Yes | 309 | 307 | 99.4 | 2 | 0.6 | |
| No | 860 | 440 | 51.2 | 420 | 48.8 | |
| Recurrence of primary complaints | | | | | | <0.001 |
| Yes | 124 | 3 | 2.4 | 121 | 97.6 | |
| No | 1045 | 744 | 71.2 | 301 | 28.8 | |
| Non-surgical treatment | | | | | | <0.001 |
| Yes | 366 | 362 | 98.9 | 4 | 1.1 | |
| No | 803 | 385 | 47.9 | 418 | 52.1 | |
| Further diagnostics | | | | | | <0.001 |
| Yes | 55 | 51 | 92.7 | 4 | 7.3 | |
| No | 1114 | 696 | 62.5 | 418 | 37.5 | |
| Post-surgical complaints | | | | | | <0.001 |
| Yes | 197 | 10 | 5.1 | 187 | 94.9 | |
| No | 972 | 737 | 75.8 | 235 | 24.2 | |
| Other complaints | | | | | | <0.001 |
| Yes | 118 | 14 | 11.9 | 104 | 88.1 | |
| No | 1051 | 733 | 69.7 | 318 | 30.3 | |
| **ICD-code at readmission** | | | | | | |
| Certain infectious and parasitic diseases, ICD: A00-B99 | | | | | | <0.001 |
| Yes | 11 | 1 | 9.1 | 10 | 90.9 | |
| No | 1158 | 746 | 64.4 | 412 | 35.6 | |
| Malignant diseases, ICD: C00-C97 | | | | | | <0.001 |
| Yes | 682 | 600 | 88.0 | 82 | 12.0 | |
| No | 487 | 147 | 30.2 | 340 | 69.8 | |
| Benign, in-situ, uncertain neoplasm, ICD: D00-D48 | | | | | | 0.042 |
| Yes | 28 | 23 | 82.1 | 5 | 17.9 | |
| No | 1141 | 724 | 63.5 | 417 | 36.5 | |
| Blood forming organ diseases, ICD: D50-D90 | | | | | | 0.002 |
| Yes | 11 | 2 | 18.2 | 9 | 81.8 | |
| No | 1158 | 745 | 64.3 | 413 | 35.7 | |
| Endocrine and metabolic diseases, ICD: E00-E90 | | | | | | 0.452 |
| Yes | 1 | 1 | 100.0 | 0 | 0.0 | |
| No | 1168 | 746 | 63.9 | 422 | 36.1 | |
| Mental and behavioral disorder, ICD: F00-F99 | | | | | | 0.270 |
| Yes | 3 | 1 | 33.3 | 2 | 66.7 | |
| No | 1166 | 746 | 64.0 | 420 | 36.0 | |
| Nervous system diseases, ICD: G00-G99 | | | | | | 0.118 |
| Yes | 44 | 33 | 75.0 | 11 | 25.0 | |
| No | 1125 | 714 | 63.5 | 411 | 36.5 | |
| Eye/ ear diseases, ICD: H00-H95 | | | | | | <0.001 |
| Yes | 67 | 19 | 28.4 | 48 | 71.6 | |

*(Continued)*

**Table 1.** (Continued)

| | | | | | | |
|---|---|---|---|---|---|---|
| No | 1102 | 728 | 66.1 | 374 | 33.9 | |
| Circulatory system diseases, ICD: I00-I99 | | | | | | 0.001 |
| Yes | 6 | 0 | 0.0 | 6 | 100.0 | |
| No | 1163 | 747 | 64.2 | 416 | 35.8 | |
| Respiratory system diseases, ICD: J00-J99 | | | | | | <0.001 |
| Yes | 86 | 32 | 37.2 | 54 | 62.8 | |
| No | 1083 | 715 | 66.0 | 368 | 34.0 | |
| Gastrointestinal tract diseases, ICD: K00-K93 | | | | | | 0.012 |
| Yes | 23 | 9 | 39.1 | 14 | 60.9 | |
| No | 1146 | 738 | 64.4 | 408 | 35.6 | |
| Skin and subcutaneous tissue diseases, ICD: L00-L99 | | | | | | 0.709 |
| Yes | 7 | 4 | 57.1 | 3 | 42.9 | |
| No | 1162 | 743 | 63.9 | 419 | 36.1 | |
| Musculoskeletal system/connective tissue diseases, ICD: M00-M99 | | | | | | 0.270 |
| Yes | 3 | 1 | 33.3 | 2 | 67.7 | |
| No | 1166 | 746 | 64.0 | 420 | 36.0 | |
| Genitourinary system diseases, ICD: N00-N99 | | | | | | NA |
| Yes | 0 | 0 | 0.0 | 0 | 0.0 | |
| No | 1169 | 747 | 63.9 | 422 | 36.1 | |
| Congenital malformations and chromosomal abnormalities, ICD: Q00-Q99 | | | | | | 0.856 |
| Yes | 5 | 3 | 60.0 | 2 | 40.0 | |
| No | 1164 | 744 | 63.9 | 420 | 36.1 | |
| Symptoms, signs, abnormal findings, ill-defined causes, not otherwise classified, ICD: R00-R99 | | | | | | <0.001 |
| Yes | 53 | 4 | 6.3 | 49 | 93.7 | |
| No | 1116 | 743 | 66.6 | 373 | 33.4 | |
| Injury, poisoning and certain other consequences of external causes, ICD: S00-T98 | | | | | | <0.001 |
| Yes | 136 | 10 | 7.4 | 126 | 92.6 | |
| No | 1033 | 737 | 71.3 | 296 | 28.7 | |
| Factors influencing good health and other utilization of the health care system, ICD: Z00-Z99 | | | | | | 0.132 |
| Yes | 4 | 4 | 4.7 | 0 | 95.3 | |
| No | 1165 | 743 | 63.8 | 422 | 36.2 | |
| **OPS-Codes, categorized** | | | | | | |
| Diagnostics | | | | | | <0.001 |
| Yes | 310 | 233 | 75.2 | 77 | 24.8 | |
| No | 859 | 514 | 59.8 | 345 | 40.2 | |
| Imaging diagnostics | | | | | | 0.374 |
| Yes | 155 | 104 | 67.1 | 51 | 32.9 | |
| No | 1014 | 643 | 63.4 | 371 | 36.6 | |
| Surgery | | | | | | 0.404 |
| Yes | 498 | 325 | 65.3 | 173 | 34.7 | |
| No | 671 | 422 | 62.9 | 249 | 37.1 | |
| Drug treatment | | | | | | <0.001 |
| Yes | 107 | 93 | 86.9 | 14 | 13.1 | |
| No | 1062 | 654 | 61.6 | 408 | 38.4 | |
| Non-surgical treatment | | | | | | <0.001 |
| Yes | 570 | 449 | 78.8 | 121 | 21.2 | |
| No | 599 | 298 | 49.7 | 301 | 50.3 | |
| Adjuvant treatment | | | | | | 0.084 |
| Yes | 49 | 37 | 75.5 | 12 | 24.5 | |
| No | 1120 | 710 | 63.4 | 410 | 36.6 | |

(*Continued*)

**Table 1.** (Continued)

| Main diagnosis identical to primary main diagnosis | | | | | | <0.001 |
|---|---|---|---|---|---|---|
| Ja | 824 | 638 | 75.8 | 204 | 24.2 | |
| Nein | 860 | 109 | 33.3 | 218 | 66.7 | |
| **PCCL at primary treatment** | | | | | | 0.001 |
| High (0–1) | 258 | 187 | 72.5 | 71 | 27.5 | |
| Low (2–4) | 661 | 398 | 60.2 | 263 | 39.8 | |
| **PCCL at readmission** | | | | | | <0.001 |
| High (0–1) | 258 | 196 | 76.0 | 62 | 24.0 | |
| Low (2–4) | 658 | 386 | 58.7 | 272 | 41.3 | |
| **DRG-partition at primary treatment** | | | | | | 0.011 |
| Surgical | 621 | 376 | 60.5 | 245 | 39.5 | |
| Medical | 540 | 366 | 67.8 | 174 | 32.2 | |
| **DRG-partition at readmission** | | | | | | <0.001 |
| Surgical | 544 | 430 | 79.0 | 114 | 21.0 | |
| Medical | 466 | 235 | 50.4 | 231 | 49.6 | |
| **Comorbidity** | | | | | | <0.001 |
| High (≥ 4 SD) | 696 | 484 | 69.5 | 212 | 30.5 | |
| Low (< 4 SD) | 473 | 263 | 55.6 | 210 | 44.4 | |
| **Number of inpatients** | | | | | | 0.057 |
| High | 234 | 137 | 58.5 | 97 | 41.5 | |
| Low | 935 | 610 | 65.2 | 325 | 34.8 | |
| **Discharge against medical advice** | | | | | | 0.018 |
| | 28 | 12 | 42.9 | 16 | 57.1 | |
| | 1135 | 733 | 64.6 | 402 | 35.4 | |
| | **Mean±SD** | **Mean±SD** | | **Mean±SD** | | |
| Age at primary treatment, years | 58.87±17.09 | 61.64±13.25 | | 53.95±21.49 | | <0.001 |
| Treatment duration at primary treatment, days | 5.68±5.62 | 5.67±6.02 | | 5.69±4.87 | | 0.968 |
| Interval between primary treatment and readmission, days | 12.51±8.20 | 13.88±7.90 | | 10.07±8.16 | | <0.001 |
| Treatment duration at readmission, days | 7.01±8.17 | 7.73±8.79 | | 5.76±6.76 | | <0.001 |
| Secondary diagnoses, n | 5.32±4.50 | 5.57±4.24 | | 4.86±4.89 | | 0.009 |

ICD = International Classification of Diseases; PCCL = Patient Clinical Complexity; SD = Secondary diagnoses.

(OR = 2.34; CI = 1.34–4.11). A main diagnosis in the ICD categories Injury, poisoning and certain other consequences of external causes (predominantly trauma cases in the analyzed otolaryngology population), ICD: S00-T98 (OR = 66.67; CI = 15.87–333.33), symptoms, signs, abnormal findings, ill-defined causes, not otherwise classified (most frequently: epistaxis, dyspnea, and vertigo), ICD: R00-R99, ICD: R00-R99 (OR = 62.5; CI = 11.76–333.33), blood forming organ diseases, ICD: D50-D90 (OR = 21.276; CI = 3.508–125), and eye/ ear diseases (predominantly ear diseases in the analyzed otolaryngology population), ICD: H00-H95 (OR = 12.66; CI = 4.29–37.03) had the highest risk for unplanned readmission.

## Discussion

This first and large population-based analysis revealed multifactorial predictors of planned and unplanned readmission for otolaryngology patients in German hospitals. The overall 30-day readmission rate was 7.6%. Surgical cases had a 30-day readmission rate of 6.5%. Readmission was planned in 4.9% and unplanned in 2.8% of the cases (Surgical cases alone: 2.6%). Former studies were mainly focused on surgical cases: Here, 30-day unplanned readmission

**Table 2. Multivariate analysis of risk factors for unplanned 30-day readmission.**

| Parameter | OR | 95% CI | p |
|---|---|---|---|
| Age, years | 0.998 | 0.983–1.013 | 0.800 |
| Treatment duration at readmission, days | 0.987 | 0.952–1.022 | 0.469 |
| Interval between primary treatment and readmission, days | 1.038 | 1.010–1.067 | 0.008 |
| Secondary diagnoses, n | 0.855 | 0.789–0.926 | <0.001 |
| **Gender** | | | 0.546 |
| Male | 1 | | |
| Female | 1.169 | 0.704–1.942 | |
| **PCCL (Primary treatment)** | | | 0.121 |
| High (0–1) | 1 | | |
| Low (2–4) | 1.672 | 0.873–3.195 | |
| **PCCL (readmission)** | | | 0.312 |
| Low (2–4) | 1 | | |
| High (0–1) | 1.439 | 0.711–2.914 | |
| **DRG-Partition (primary treatment)** | | | 0.386 |
| Medical | 1 | | |
| Surgical | 1.259 | 0.747–2.123 | |
| **DRG-Partition (readmission)** | | | <0.001 |
| Surgical | 1 | | |
| Medical | 3.916 | 2.243–6.836 | |
| **Demission against medical advice** | | | 0.001 |
| No | 1 | | |
| Yes | 9.615 | 2.688–34.482 | |
| **Comorbidity** | | | 0.101 |
| High ($\geq 4$ SD) | 1 | | |
| Low ($< 4$ SD) | 1.740 | 0.898–3.369 | |
| **Main diagnosis identical** | | | 0.205 |
| Yes | 1 | | |
| No | 1.447 | 0.818–2.560 | |
| **ICD-codes** | | | |
| Certain infectious and parasitic diseases, ICD: A00-B99 | | | 0.048 |
| No | 1 | | |
| Yes | 10.638 | 0.009–0.981 | |
| Malignant diseases, ICD: C00-C97 | | | 0.613 |
| Yes | 1 | | |
| No | 1.270 | 0.503–3.206 | |
| Benign, in-situ, uncertain neoplasm, ICD: D00-D48 | | | 0.140 |
| No | 1 | | |
| Yes | 3.040 | 0.694–13.333 | |
| Blood forming organ diseases, ICD: D50-D90 | | | 0.001 |
| No | 1 | | |
| Yes | 21.276 | 3.508–125 | |
| Eye/ ear diseases, ICD: H00-H95 | | | <0.001 |
| No | 1 | | |
| Yes | 12.658 | 4.292–37.037 | |
| Circulatory system diseases, ICD: I00-I99 | | | 0.999 |
| Yes | 1 | | |
| No | NA | NA | |

(*Continued*)

**Table 2.** (Continued)

| Parameter | OR | 95% CI | p |
|---|---|---|---|
| Respiratory system diseases, ICD: J00-J99 | | | <0.001 |
| No | 1 | | |
| Yes | 5.814 | 2.217–15.385 | |
| Gastrointestinal tract diseases, ICD: K00-K93 | | | 0.020 |
| No | 1 | | |
| Yes | 4.831 | 1.277–18.181 | |
| Symptoms, signs, abnormal findings, ill-defined causes, not otherwise classified, ICD: R00-R99 | | | <0.001 |
| No | 1 | | |
| Yes | 62.5 | 11.765–333.333 | |
| Injury, poisoning and certain other consequences of external causes, ICD: S00-T98 | | | <0.001 |
| No | 1 | | |
| Yes | 66.667 | 15.873–333.333 | |
| **OPS-Code, categorized** | | | |
| **Diagnostics** | | | 0.003 |
| Yes | 1 | | |
| No | 2.343 | 1.337–4.105 | |
| **Drug treatment** | | | 0.062 |
| Yes | 1 | | |
| No | 2.455 | 0.956–6.302 | |
| **Surgical treatment** | | | <0.001 |
| No | 1 | | |
| Yes | 3.330 | 1.862–5.957 | |

OR = Odds ratio, CI = confidence interval; ICD = International Classification of Diseases; PCCL = Patient Clinical Complexity; SD = Secondary diagnoses.

rates vary from 5% to 20% [14–17]. The highest rates in otolaryngology patients are reported for head and neck cancer patients, especially following laryngectomy [15, 18, 19]. Specialty readmission rates are typically lower in surgical departments (about 10–12%) than in internal medicine (about 20%) [20, 21]. Unplanned 30-day readmission rates in general surgery reach maximally 10% in newer studies after implementation of HRRP strategies [22]. It can be concluded that the unplanned 30-day readmission rate was low compared to other otolaryngology studies and especially lower compared to other surgical disciples and much lower than reported for non-surgical disciplines.

The detected readmission predictors 1) head and neck cancer, 2) higher comorbidity, and 3) non-surgical cases have also been shown to be relevant for US-American otolaryngology patients treated following the introduction of healthcare quality programs like the National Surgical Quality Improvement Program (NSQIP) [6, 9, 14–16]. High-volume departments had also a higher overall readmission rate. This seems to be to a general association seen in many hospitals independently from a specialty [20], but the results in the literature are controversial [16]. Because not only head and neck cancer patients like in most previous studies but all kind of otolaryngology patients were included in the present study, it could be shown that also (in descending order) otoneurological diseases (mainly acute vestibular syndrome; acute hearing

loss), infectious diseases (mainly erysipelas, herpes zoster), and musculoskeletal system diseases (mainly acquired outer ear deformities) had a high association (OR>3) for readmission.

Discharge against medical advice seems to be a self-explanatory risk factor, seems to be an important risk factor in countries with high amount of patients with healthcare insurance [23], and was analyzed so far only in few population-based studies in countries with DRG or comparable healthcare financing systems [24]. The ICD coding of bleeding in the airways and coagulopathies is mainly found in patients with recurrent epistaxis. Epistaxis is a well-known and a factor difficult to control to prevent unplanned readmission [25]. In contrast to planned readmission, head and neck cancer was not an independent risk factor for unplanned readmission. Nevertheless, the 2485 patients with head and neck cancer formed an important subgroup. 1016 of the patients with head and neck cancer underwent head and neck surgery. In a recent study focusing on 660 US-American patients who underwent head and neck surgery, length of stay at the index admission longer than 5 days was a strong predicator for unplanned 30-day admission [17]. On the other hand, wound infection was the most common reason for 30-day admission in the US-American study. The authors speculate that keeping the patients in the hospital longer may have prevented the unplanned readmission. However, a longer length of stay could have contributed to the infection. The mean length of stay in the US-American study was 5.6 days. This is much shorter than in the present study. The mean length of stay for the German head and neck cancer patients was 9.0 days. This may be attributed to different health practices in both countries. In the end, it may be speculated that the higher rate of planned 30-day readmission lead to less unplanned 30-day admissions in the German head and neck cancer patients. Also in contrast to a general higher readmission rate, high-volume hospitals seem not to have a higher rate of unplanned readmissions [21]. This can be confirmed for otolaryngology departments. After multivariate analysis, high-volume otolaryngology departments did not have a higher rate of unplanned readmissions.

The three most frequent single diagnoses at index diagnoses (apart from already discussed patients with epistaxis as well as head and neck cancer cases were: chronic tonsillitis (ICD: J35), unspecified disorders of nose and paranasal sinuses (ICD: J34), and chronic sinusitis (ICD: J32). 98% of the patients were surgical cases. In all but one of these unplanned readmissions, the reason was secondary bleeding or wound infection. Most of the surgical site bleeding complications occurred later than 7 days after discharge. Hence, probably most of the readmissions because of bleeding were not predictable and preventable. Unfortunately, the perioperative antibiotic treatment and the discharge antibiotics were not recorded. Discharge antibiotics were not preventive of infection and readmission in another study [17]. Furthermore, a wider use of prophylactic antibiotics can definitely be viewed critically.

The present study has several limitations and strengths. Due to the retrospective design, several parameters with possible association to unplanned readmission could not be analyzed due to a lack of sufficient data. For instance, in-hospital complications during the index admission seem to be a very strong predictor associated with 30-day readmission [6, 9, 14–16]. Furthermore, in head and neck cancer patients, cancer subsite, type of procedure, socioeconomic factors, comorbidities like coronary artery disease, chronic renal failure, or presence of a gastrostomy tube influence the risk of readmission [14, 16, 17]. A detailed analysis of cancer subsites and procedures was beyond the aim of the present more general overview on otolaryngology patients. Administrative data may not reliably describe the reason for readmission, especially when the rate of unplanned readmission should be analyzed [26]. A strength of the present study is that all charts of readmitted patients were examined individually, rather than relying on coding data alone.

Not much is known about strategies to prevent unplanned readmission in otolaryngology patients beyond HRRP or comparable policies. Prospective trials are lacking. We identified

only one prospective trial, showing that perioperative education programs for patients and caregivers (pre-operative hands-on classes, booklets about the treatment journey, discharge coaching) seem to be an effective approach to reducing unplanned readmission in head and neck cancer patients [15]. Reducing care fragmentation after discharge might be another strategy, at least in head and neck cancer patients, to reduce unplanned readmissions [27]. Machine learning algorithms to predict the individual 30-day readmission probability may be at least a future option to better identify patients at risk much earlier [28].

## Conclusions

The most important factors for 30-day unplanned readmission were: 1) discharge due to patient's request against medical advice, 2) surgical cases, and 3) several ICD categories. Injury, poisoning and certain other consequences of external causes (mainly unspecified complications), symptoms, signs, abnormal findings, ill-defined causes, not otherwise classified, ICD: R00-R99 (mainly bleeding in the airways and dysphagia), blood forming organ diseases, ICD: D50-D90 (mainly coagulopathies), and eye/ ear diseases (mainly: acute vestibular syndrome, acute hearing loss) were associated with higher risk for unplanned readmission. This should allow developing concepts to reduce the rate of unplanned readmissions in the German DRG-system for safer and better otolaryngology inpatient care. Patients with the mentioned risk factors (and caregivers) should receive specific perioperative education programs addressing the related reasons that might lead to readmission. Prophylactic measures like prophylactic treatment if appropriate or a higher frequency of follow-up visits in the hand of one responsible physician might help to reduce the risk of unplanned 30-day readmission.

## Supporting information

**S1 Table. Overview about all patients: 16,444 inpatient cases of 12,859 patients in 2015.**
(DOCX)

**S2 Table. Comparison of the group of patients with 30-day readmission to the group of patients without readmission.**
(DOCX)

**S3 Table. Multivariate analysis of risk factors for 30-day readmission.**
(DOCX)

**S1 Fig. Localization of the primary disease of all patients at primary index admission.**
(DOCX)

**S2 Fig. The hundred most frequent primary ICD codes of all patients at primary index admission.** A: First 50 most frequent ICD codes. B: Second 50 most frequent ICD codes.
(DOCX)

## Acknowledgments

We would like to thank the staff of medical controlling in the seven Thuringian hospitals for the help for the help to collect the data.

## Author Contributions

**Conceptualization:** Orlando Guntinas-Lichius.

**Data curation:** Wido Rippe, Andreas Dittberner, Daniel Boeger, Jens Buentzel, Kerstin Hoffmann, Holger Kaftan, Andreas Mueller, Gerald Radtke, Orlando Guntinas-Lichius.

**Formal analysis:** Wido Rippe, Andreas Dittberner, Orlando Guntinas-Lichius.

**Investigation:** Wido Rippe, Andreas Dittberner, Daniel Boeger, Jens Buentzel, Kerstin Hoffmann, Holger Kaftan, Andreas Mueller, Gerald Radtke.

**Methodology:** Wido Rippe, Andreas Dittberner, Orlando Guntinas-Lichius.

**Project administration:** Orlando Guntinas-Lichius.

**Resources:** Jens Buentzel, Kerstin Hoffmann, Holger Kaftan, Gerald Radtke, Orlando Guntinas-Lichius.

**Software:** Orlando Guntinas-Lichius.

**Validation:** Orlando Guntinas-Lichius.

**Visualization:** Orlando Guntinas-Lichius.

**Writing – original draft:** Orlando Guntinas-Lichius.

**Writing – review & editing:** Andreas Dittberner, Daniel Boeger, Jens Buentzel, Kerstin Hoffmann, Holger Kaftan, Andreas Mueller, Gerald Radtke, Orlando Guntinas-Lichius.

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
