## [Decision Letter · Decision Letter 0]

13 Aug 2019

PONE-D-19-21224

30-day unplanned readmission rate in otolaryngology patients: a population-based study in Thuringia, Germany

PLOS ONE

Dear Prof Guntinas-Lichius,

Thank you for submitting your manuscript to PLOS ONE. After careful consideration, we feel that it has merit but does not fully meet PLOS ONE’s publication criteria as it currently stands. Therefore, we invite you to submit a revised version of the manuscript that addresses the points raised during the review process.

We would appreciate receiving your revised manuscript by Sep 27 2019 11:59PM. To enhance the reproducibility of your results, we recommend that if applicable you deposit your laboratory protocols in protocols.io, where a protocol can be assigned its own identifier (DOI) such that it can be cited independently in the future. For instructions see: http://journals.plos.org/plosone/s/submission-guidelines#loc-laboratory-protocols

We look forward to receiving your revised manuscript.

Kind regards,

Peter Dziegielewski, MD, FRCSC

Academic Editor

PLOS ONE

2. Thank you for including your ethics statement: The Ethics Committee waived the requirement for informed consent for the patients with exclusive retrospective data analysis using the patients’ charts.

Reviewers' comments:

Reviewer's Responses to Questions

**Comments to the Author**

1. Is the manuscript technically sound, and do the data support the conclusions?

Reviewer #1: Yes

Reviewer #2: Yes

Reviewer #3: Partly

2. Has the statistical analysis been performed appropriately and rigorously? 

Reviewer #1: Yes

Reviewer #2: Yes

Reviewer #3: I Don't Know

3. Have the authors made all data underlying the findings in their manuscript fully available?

Reviewer #1: Yes

Reviewer #2: Yes

Reviewer #3: No

4. Is the manuscript presented in an intelligible fashion and written in standard English?

Reviewer #1: Yes

Reviewer #2: Yes

Reviewer #3: No

5. Review Comments to the Author

Reviewer #1: The authors investigate a heterogeneous group of otolaryngology patients across multiple institutions in Germany. The concept to identify predictors for unplanned readmissions is important. It is a bit challenging to extrapolate further data from ICD codes, leaving the findings a bit unfocused and vague, although this is the nature of the data. Centering the discussion and elaborating on the most interesting findings, with extended discussion on impact and opportunities for quality improvement would strengthen the manuscript. Some efforts to clarify and investigate the manuscript further would be helpful in enhancing its value.

Some additional points:

1. Can authors clarify how they defined “planned” readmission, perhaps give some examples? It is unclear in the manuscript what a “planned” readmission is.

2. The authors should edit for grammar throughout the manuscript. e.g. pg. 4 lines 104-105 "have be performed" and Table I the use of "ja" and "nein".

3. Figure 2 image quality could be improved.

Reviewer #2: GENERAL COMMENTS

Rippe and colleagues provide a study of unplanned readmissions in otolaryngology patients highlighting several medical conditions and demographic factors predictive of patient readmission. The study is well justified in terms of potential clinical and economic impact. As the authors point out, existing studies in this area are limited and narrowly focused. The authors were able to obtain a large sample size, and the methodological approaches, analyses, and conclusions all seem appropriate. The study will be a welcome addition to the literature database accessed by practitioners and researchers in otolaryngology. My feedback is minimal and limited to suggestions for improved clarity.

SPECIFIC COMMENTS

Major

In its current format, Figure 2 is not useful for readers without a computer nearby to lookup ICD codes. Even these readers are burdened with the task of cross referencing the codes to the conditions, assuming they haven’t memorized them all. A brief text description would greatly facilitate understanding the main points of this figure, and can easily be added, e.g., by switching the bar graph orientation from horizontal to vertical.

Another feature that would be helpful for Figure 2 is a way to directly compare ICDs common for planned vs unplanned readmissions. For instance, the grouped bar plots in Figure 1 make it easy for readers to identify disease locations associated with planned (e.g., pharynx, larynx, neck) vs unplanned (e.g., ear, paranasal sinus, nose). There are several ways this type of comparison could be implemented in Figure 2, for instance, separate grouped or stacked bar plots ranking by planned and unplanned readmissions.

Minor

146: Please provide additional detail to disambiguate the term “multivariable analysis”. Does this imply multivariate regression? Multivariate analysis of variance?

195-197: “but re-surgery because of postoperative complaints occurred more often unplanned (p<0.001).”

It’s clear from Table 1 that Further surgery was more common for Planned readmission, and that Recurrence of primary complaints was more common for Unplanned readmission. These independent observations notwithstanding, I do not see anything supporting the statement that “re-surgery” (use presumably synonymous with further surgery) was performed specifically because of postoperative complaints.

Editorial

73: ICD -> International Classification of Diseases (ICD)

98: 30-day readmission rate as parameter -> 30-day readmission rate as a parameter

104-106: Only a few studies and mainly based on the experience in a single hospital have be performed to analyze risk factors for unplanned hospital readmission in otolaryngology patients [6-8]. -> Only a few studies, which are mainly based on the experience in a single hospital, have be performed to analyze risk factors for unplanned hospital readmission in otolaryngology patients [6-8].

113: (for instance, [10-13]. -> (for instance, [10-13].)

146: multivariable analysis -> multivariate analysis

165: Due to the multivariate analysis -> According to the multivariate analysis

172: 30day -> 30-day

195: Further surgery took place rather planned than unplanned -> Further surgery took place more often for planned than unplanned

196: re-surgery -> further surgery

197: origin complaints -> original complaints

312-313: A strength of the present study is that all charts of readmitted patients were examined individually. It was not just relied on coding data. -> A strength of the present study is that all charts of readmitted patients were examined individually, rather than relying on coding data alone.

318: a potent measure to reduce unplanned readmission -> an effective approach to reducing unplanned readmission

Reviewer #3: This exploratory study aimed to "analyze associations between patients’ characteristics and treatment factors with 30 day unplanned readmissions in hospitalized otolaryngology patients". It then proceeded to present a number of possible risk factors for readmission based on the results of the univariate and multivariate analysis. Some of the risk factors that had significant associations appear counter intuitive. For example, while older patients, those with malignant disease and those with higher co morbidity had a higher risk of readmission, they also found that non-surgical cases had a higher risk of readmission than surgical cases.

They found out that unplanned readmissions were more frequent among those who were discharged against medical advice. They then listed several ICD codes that appeared more frequently among those with unplanned readmissions. None of these were explained.

I think the authors should have first defined what is a planned and an unplanned readmission.

I think that the authors should have made clear from the start what are these associations that they planned to analyze and why. I would have appreciated an a priori explanatory framework relating the most important putative factors to the outcome (i.e. unplanned readmission) which would have guided them in the analysis. The framework would have also presented the biological or clinical rationale for focusing on these putative risk factors. Because there was no explanatory framework they could not explain the results that they obtained.

The tables need to be shortened to what are essential. For example, the p values for the ICD codes which do not appear to be clinically related to otolaryngologic outcomes may probably need to be deleted.

The discussion part repeated what was found in the results and even added information that might better have been placed in the results section.

The conclusion does not recapitulate the most important findings and how they apply to improving the quality of care for otolaryngologic patients.

6. PLOS authors have the option to publish the peer review history of their article (what does this mean?). If published, this will include your full peer review and any attached files.

Reviewer #1: No

Reviewer #2: No

Reviewer #3: Yes: Dr. Jose M. Acuin

---

## [Author Response · Author response to Decision Letter 0]

27 Aug 2019

Rebuttal letter

PONE-D-19-21224

30-day unplanned readmission rate in otolaryngology patients: a population-based study in Thuringia, Germany

Thank you very much for the detailed reviews. Herewith, we would like to respond to all queries.

1. Editorial comments

1.1. Thank you for including your ethics statement: The Ethics Committee waived the requirement for informed consent for the patients with exclusive retrospective data analysis using the patients’ charts. Please amend your current ethics statement to include the full name of the ethics committee/institutional review board(s) that approved your specific study. Once you have amended this/these statement(s) in the Methods section of the manuscript, please add the same text to the “Ethics Statement” field of the submission form (via “Edit Submission”).

Answer 1.1) Done. We added the full name of the ethics committee and added the same text in the submission form.

1.2. In your revised cover letter, please address the following prompts: a) If there are ethical or legal restrictions on sharing a de-identified data set, please explain them in detail (e.g., data contain potentially identifying or sensitive patient information) and who has imposed them (e.g., an ethics committee). Please also provide contact information for a data access committee, ethics committee, or other institutional body to which data requests may be sent. b) If there are no restrictions, please upload the minimal anonymized data set necessary to replicate your study findings as either Supporting Information files or to a stable, public repository and provide us with the relevant URLs, DOIs, or accession numbers. Please see http://www.bmj.com/content/340/bmj.c181.long for guidelines on how to de-identify and prepare clinical data for publication. For a list of acceptable repositories, please see http://journals.plos.org/plosone/s/data-availability#loc-recommended-repositories. We will update your Data Availability statement on your behalf to reflect the information you provide.

Answer 1.2) Done. We added this information in the cover letter.

2. Reviewer #1

2.1 The authors investigate a heterogeneous group of otolaryngology patients across multiple institutions in Germany. The concept to identify predictors for unplanned readmissions is important. It is a bit challenging to extrapolate further data from ICD codes, leaving the findings a bit unfocused and vague, although this is the nature of the data. Centering the discussion and elaborating on the most interesting findings, with extended discussion on impact and opportunities for quality improvement would strengthen the manuscript. Some efforts to clarify and investigate the manuscript further would be helpful in enhancing its value.

Answer 2.1) We followed the suggestions of all three reviewers. The result is: We focused the text by giving more summarizing information on the figures in the Results. The description of the ICD codes makes it much easier now to understand the disease behind the codes. We deleted parts of repeated results in the Discussion. We clarified definitions and aims.

Some additional points

2.2. Can authors clarify how they defined “planned” readmission, perhaps give some examples? It is unclear in the manuscript what a “planned” readmission is.

Answer 2.2) Done. We added the following sentences at the end of the Method sections: “. A planned readmission was defined as a readmission that was planned during the index admission. These patients received an appointment for readmission at discharge from the hospital. All other readmissions were defined as unplanned readmissions.”.

2.3. The authors should edit for grammar throughout the manuscript. e.g. pg. 4 lines 104-105 "have be performed" and Table I the use of "ja" and "nein".

Answer 2.3) We checked the manuscript completely and corrected the mentioned two errors.

2.4. Figure 2 image quality could be improved.

Answer 2.4) Done. We changed the format completely, following the suggestion of reviewer #2, see answer 3.2.

3. Reviewer #2

3.1. Rippe and colleagues provide a study of unplanned readmissions in otolaryngology patients highlighting several medical conditions and demographic factors predictive of patient readmission. The study is well justified in terms of potential clinical and economic impact. As the authors point out, existing studies in this area are limited and narrowly focused. The authors were able to obtain a large sample size, and the methodological approaches, analyses, and conclusions all seem appropriate. The study will be a welcome addition to the literature database accessed by practitioners and researchers in otolaryngology. My feedback is minimal and limited to suggestions for improved clarity.

Answer 3.1) Thanks.

Major

3.2. In its current format, Figure 2 is not useful for readers without a computer nearby to lookup ICD codes. Even these readers are burdened with the task of cross referencing the codes to the conditions, assuming they haven’t memorized them all. A brief text description would greatly facilitate understanding the main points of this figure, and can easily be added, e.g., by switching the bar graph orientation from horizontal to vertical. Another feature that would be helpful for Figure 2 is a way to directly compare ICDs common for planned vs unplanned readmissions. For instance, the grouped bar plots in Figure 1 make it easy for readers to identify disease locations associated with planned (e.g., pharynx, larynx, neck) vs unplanned (e.g., ear, paranasal sinus, nose). There are several ways this type of comparison could be implemented in Figure 2, for instance, separate grouped or stacked bar plots ranking by planned and unplanned readmissions.

Answer 3.2) We re-arranged Figure 2. We switched the orientation of the bars and added text to the ICD codes

Minor

3.3. 146: Please provide additional detail to disambiguate the term “multivariable analysis”. Does this imply multivariate regression? Multivariate analysis of variance?

Answer 3.2) See also answer 3.9. This sentence is leading to the next sentence explaining what is meant by multivariate analysis. To make this clear we changed the full stop to a colon: “… in the multivariate analysis: Binary logistic regression analysis was used to determine …”.

3.4. 195-197: “but re-surgery because of postoperative complaints occurred more often unplanned (p<0.001).”

It’s clear from Table 1 that Further surgery was more common for Planned readmission, and that Recurrence of primary complaints was more common for Unplanned readmission. These independent observations notwithstanding, I do not see anything supporting the statement that “re-surgery” (use presumably synonymous with further surgery) was performed specifically because of postoperative complaints.

Answer 3.4) Yes. Further surgery is meant. We changed this. Re-surgery is misleading. See also answer 3.13.

Editorial

3.5. 73: ICD -> International Classification of Diseases (ICD)

Answer 3.5) Changed.

3.6. 98: 30-day readmission rate as parameter -> 30-day readmission rate as a parameter

Answer 3.6) Changed.

3.7. 104-106: Only a few studies and mainly based on the experience in a single hospital have be performed to analyze risk factors for unplanned hospital readmission in otolaryngology patients [6-8]. -> Only a few studies, which are mainly based on the experience in a single hospital, have be performed to analyze risk factors for unplanned hospital readmission in otolaryngology patients [6-8].

Answer 3.7) Changed.

3.8. 113: (for instance, [10-13]. -> (for instance, [10-13].)

Answer 3.8) Changed.

3.9. 146: multivariable analysis -> multivariate analysis

Answer 3.9) Changed.

3.10. 165: Due to the multivariate analysis -> According to the multivariate analysis

Answer 3.10) Changed.

3.11. 172: 30day -> 30-day

Answer 3.11) Changed.

3.12. 195: Further surgery took place rather planned than unplanned -> Further surgery took place more often for planned than unplanned

Answer 3.12) Changed.

3.13. 196: re-surgery -> further surgery

Answer 3.13) Changed.

3.14. 197: origin complaints -> original complaints

Answer 3.14) Changed.

3.15. 312-313: A strength of the present study is that all charts of readmitted patients were examined individually. It was not just relied on coding data. -> A strength of the present study is that all charts of readmitted patients were examined individually, rather than relying on coding data alone.

Answer 3.15) Changed

3.16. 318: a potent measure to reduce unplanned readmission -> an effective approach to reducing unplanned readmission

Answer 3.16) Changed.

4. Reviewer #3

4.1. This exploratory study aimed to "analyze associations between patients’ characteristics and treatment factors with 30 day unplanned readmissions in hospitalized otolaryngology patients". It then proceeded to present a number of possible risk factors for readmission based on the results of the univariate and multivariate analysis. Some of the risk factors that had significant associations appear counter intuitive. For example, while older patients, those with malignant disease and those with higher co morbidity had a higher risk of readmission, they also found that non-surgical cases had a higher risk of readmission than surgical cases.

Answer 4.1) No comment needed.

4.2. They found out that unplanned readmissions were more frequent among those who were discharged against medical advice. They then listed several ICD codes that appeared more frequently among those with unplanned readmissions. None of these were explained.

Answer 4.2) We added several sentences in the Results to make is easier to understand the diseases behind the ICD codes. Furthermore, see also answer 3.2, we re-arranged figure 2, so that the ICD codes are explained now.

4.3. I think the authors should have first defined what is a planned and an unplanned readmission.

Answer 4.3 Done. See answer 2.2.

4.4. I think that the authors should have made clear from the start what are these associations that they planned to analyze and why. I would have appreciated an a priori explanatory framework relating the most important putative factors to the outcome (i.e. unplanned readmission) which would have guided them in the analysis. The framework would have also presented the biological or clinical rationale for focusing on these putative risk factors. Because there was no explanatory framework they could not explain the results that they obtained.

Answer 4.4.) We hope that we understood the direction of this comment. We added in the Methods at the end of the final paragraph: “The primary aim of the subsequent evaluation was to analyze associations between patients’ characteristics and treatment factors with 30-day unplanned readmissions in hospitalized otolaryngology patients in the German Diagnosis Related Group (D-DRG) system. The G-DRG system already includes instruments to avoid early readmission, especially by sanctioning the reimbursement. We hypothesized that, nevertheless, specific otolaryngology diseases and type of treatment as well as patients’ comorbidity have influence on the risk of 30-day unplanned readmission.”.

4.5. The tables need to be shortened to what are essential. For example, the p values for the ICD codes which do not appear to be clinically related to otolaryngologic outcomes may probably need to be deleted.

Answer 4.5) It is policy of PLOS one (as it is nowadays for several journals) to disclose as many of the data as possible. Furthermore, we think, that it is also important for the reader to understand which factors are not important (not significant in statistical analysis). Therefore, we would not like to delete parameters from the otherwise complete tables. To make Table 1 more readable, we split table 1 into table 1A and table 1B.

4.6. The discussion part repeated what was found in the results and even added information that might better have been placed in the results section.

Answer 4.6) We shifted sentences summarizing results from the Discussion (the Discussion already was not very long) back to the Results. We tried now to reduce results in the Discussion to information needed to compare the data to other studies. 

4.7. The conclusion does not recapitulate the most important findings and how they apply to improving the quality of care for otolaryngologic patients.

Answer 4.7) The Conclusion was revised. We recapitulate now the most important findings. And we present now some evident strategies to reduce unplanned readmissions-

Orlando Guntinas-Lichius

for all authors

Jena, 20-August-2019

---

## [Decision Letter · Decision Letter 1]

12 Sep 2019

[EXSCINDED]

PONE-D-19-21224R1

30-day unplanned readmission rate in otolaryngology patients: a population-based study in Thuringia, Germany

PLOS ONE

Dear Prof Guntinas-Lichius,

Thank you for submitting your manuscript to PLOS ONE. After careful consideration, we feel that it has merit but does not fully meet PLOS ONE’s publication criteria as it currently stands. Therefore, we invite you to submit a revised version of the manuscript that addresses the points raised during the review process.

We would appreciate receiving your revised manuscript by Oct 27 2019 11:59PM. To enhance the reproducibility of your results, we recommend that if applicable you deposit your laboratory protocols in protocols.io, where a protocol can be assigned its own identifier (DOI) such that it can be cited independently in the future. For instructions see: http://journals.plos.org/plosone/s/submission-guidelines#loc-laboratory-protocols

We look forward to receiving your revised manuscript.

Kind regards,

Peter Dziegielewski, MD, FRCSC

Academic Editor

PLOS ONE

Additional Editor Comments (if provided):

Thank you to the authors for addressing most of the reviewers comments. There are still a few outstanding items which need to be answered (see reviewer comments).

Also:

1. Please give a concrete example of "planned re-admission".

2. Please double check the tables to ensure that no labels are missing. Some rows/columns are blank.

3. Please discuss risk factors for readmission in more detail.

4. Please address how head and neck surgery patients may have affected the results. There is a paper (Dziegielewski et al 2016) that discuss this group of patients in detail. It would be worthwhile comparing results.

Reviewers' comments:

Reviewer's Responses to Questions

**Comments to the Author**

1. If the authors have adequately addressed your comments raised in a previous round of review and you feel that this manuscript is now acceptable for publication, you may indicate that here to bypass the “Comments to the Author” section, enter your conflict of interest statement in the “Confidential to Editor” section, and submit your "Accept" recommendation.

Reviewer #1: (No Response)

Reviewer #2: All comments have been addressed

Reviewer #3: All comments have been addressed

2. Is the manuscript technically sound, and do the data support the conclusions?

Reviewer #1: Yes

Reviewer #2: Yes

Reviewer #3: Yes

3. Has the statistical analysis been performed appropriately and rigorously? 

Reviewer #1: Yes

Reviewer #2: Yes

Reviewer #3: Yes

4. Have the authors made all data underlying the findings in their manuscript fully available?

Reviewer #1: Yes

Reviewer #2: Yes

Reviewer #3: Yes

5. Is the manuscript presented in an intelligible fashion and written in standard English?

Reviewer #1: Yes

Reviewer #2: Yes

Reviewer #3: No

6. Review Comments to the Author

Reviewer #1: Overall the authors have improved the manuscript. I am still confused as to the concept of a "planned readmission" before discharge from the hospital. In what clinical scenarios would this happen? For a planned reconstruction surgery? For a second stage procedure? Can the authors provide an example of "planned readmission"? Otherwise other concerns have been addressed.

Reviewer #2: The authors have adequately addressed each of the suggested revisions. My only remaining suggestion is to add x-axis labels

to Figure 2 (e.g., "Number of cases" as in Figure 1).

Reviewer #3: Please correct all typographical errors (e.g., punctuation, capitalizatio of common nouns, etc). In the conclusions section, the authors may also want to state how the findings can be used to improve hospital discharge policies.

7. PLOS authors have the option to publish the peer review history of their article (what does this mean?). If published, this will include your full peer review and any attached files.

Reviewer #1: No

Reviewer #2: No

Reviewer #3: Yes: Jose M. Acuin

---

## [Author Response · Author response to Decision Letter 1]

23 Sep 2019

Rebuttal letter

PONE-D-19-21224

30-day unplanned readmission rate in otolaryngology patients: a population-based study in Thuringia, Germany

Thank you very much for the second round of detailed reviews. Herewith, we would like to respond to all queries.

1. Editorial comments

1.1. Please give a concrete example of "planned re-admission".

Answer 1.1. We added some sentences for definition in the Methods on page 5: “Planned readmissions were defined as readmissions that were planned and agreed during the index admission. For instance, a patient with head and neck cancer had his index admission for staging and panendoscopy. Typically, if surgery was planned for definitive treatment, the day of readmission already was scheduled before demission. All other patients without planned readmission were defined as unplanned readmissions, for instance when a patient was readmitted for a complication after surgery.”.

1.2. Please double check the tables to ensure that no labels are missing. Some rows/columns are blank.

Answer 1.2. We checked all tables and supplement tables. There is no complete blank row or columns. The blank fields in many rows/columns are correct.

1.3. Please discuss risk factors for readmission in more detail.

Answer 1.3. See also answer 1.4. Including the paper of Dziegielewski et al 2016 already helped to discuss the risk factors in more detail. Furthermore, we added a paragraph in the Discussion on page 16/17 and discussed the most frequent reasons for unplanned readmission.

1.4. Please address how head and neck surgery patients may have affected the results. There is a paper (Dziegielewski et al 2016) that discuss this group of patients in detail. It would be worthwhile comparing results.

Answer 1.4. Thank you for this hint. The article of Dziegielewski et al. 2016 is now the new reference No. 17. The Discussion contains now a part on page 16 where we discuss this study in relation to our results. And we address this study in the Discussion on page 17 where we talk about the limitations of our study.

2. Reviewer #1

2.1. Overall the authors have improved the manuscript. I am still confused as to the concept of a "planned readmission" before discharge from the hospital. In what clinical scenarios would this happen? For a planned reconstruction surgery? For a second stage procedure? Can the authors provide an example of "planned readmission"? Otherwise other concerns have been addressed.

Answer 2.1. See also Answer 1.1. We added some sentences for the definition of planned and unplanned surgery on page 5 in the Methods.

3. Reviewer #2

3.1. The authors have adequately addressed each of the suggested revisions. My only remaining suggestion is to add x-axis labels to Figure 2 (e.g., "Number of cases" as in Figure 1).

Answer 3.1. Done. The x-axis in Figure 2 has now also a labeling: “Number of cases”.

4. Reviewer #3

4.1. Please correct all typographical errors (e.g., punctuation, capitalizatio of common nouns, etc). In the conclusions section, the authors may also want to state how the findings can be used to improve hospital discharge policies

Answer 4.1. The text was checked for typographical errors again. Concerning the improvement of the hospital discharge policies, the Conclusions contain three sentences: “Patients with the mentioned risk factors (and caregivers) should receive specific perioperative education programs addressing the related reasons that might lead to readmission. Prophylactic measures like prophylactic treatment if appropriate or a higher frequency of follow-up visits in the hand of one responsible physician might help to reduce the risk of unplanned 30-day readmission.”.

Orlando Guntinas-Lichius

for all authors

Jena, 22-September-2019

---

## [Editor Report · Decision Letter 2]

8 Oct 2019

30-day unplanned readmission rate in otolaryngology patients: a population-based study in Thuringia, Germany

PONE-D-19-21224R2

Dear Dr. Guntinas-Lichius,

We are pleased to inform you that your manuscript has been judged scientifically suitable for publication and will be formally accepted for publication once it complies with all outstanding technical requirements.

With kind regards,

Peter Dziegielewski, MD, FRCSC

Academic Editor

PLOS ONE
---

## [Editor Report · Acceptance letter]

10 Oct 2019

PONE-D-19-21224R2 

30-day unplanned readmission rate in otolaryngology patients: a population-based study in Thuringia, Germany 

Dear Dr. Guntinas-Lichius:

I am pleased to inform you that your manuscript has been deemed suitable for publication in PLOS ONE. Congratulations! Your manuscript is now with our production department. 

With kind regards,

on behalf of

Dr. Peter Dziegielewski 

Academic Editor

PLOS ONE